# Viability of Deficit Irrigation Pre-Exposure in Adapting Robusta Coffee to Drought Stress

Godfrey Sseremba [1,2,*], Pangirayi Bernard Tongoona [2], Pascal Musoli [1], John Saviour Yaw Eleblu [2],
Leander Dede Melomey [2], Daphne Nyachaki Bitalo [1], Evans Atwijukire [1], Joseph Mulindwa [1],
Naome Aryatwijuka [1,3], Edgar Muhumuza [1], Judith Kobusinge [1], Betty Magambo [1], Godfrey Hubby Kagezi [1],
Eric Yirenkyi Danquah [2], Elizabeth Balyejusa Kizito [3], Gerald Kyalo [4], Emmanuel Iyamulemye [4]
and Geofrey Arinaitwe [1]

1. National Coffee Research Institute, National Agricultural Research Organisation, Mukono P.O. Box 185, Uganda
2. West Africa Centre for Crop Improvement, University of Ghana Legon, Accra PMB LG30, Ghana
3. Department of Agriculture, Faculty of Agricultural Science, Uganda Christian University, Mukono P.O. Box 4, Uganda
4. Uganda Coffee Development Authority, Kampala P.O. Box 7267, Uganda
* Correspondence: gsseremba16@gmail.com

**Abstract:** *Coffea canephora* has high but inadequately exploited genetic diversity. This diversity, if well exploited, can sustain coffee productivity amidst climate change effects. Drought and heat stress are major global threats to coffee productivity, quality, and tradable volumes. It is not well understood if there is a selectable variation for drought stress tolerance in Robusta coffee half-sibs as a result of watering deficit pre-exposure at the germination stage. Half-sib seeds from selected commercial clones (KR5, KR6, KR7) and a pipeline clone X1 were primed with deficit watering at two growth stages followed by recovery and later evaluated for tolerance to watering deficit stress in three different temperature environments by estimation of plant growth and wilt parameters. Overall, the KR7 family performed the best in terms of the number of individuals excelling for tolerance to deficit watering. In order of decreasing tolerance, the 10 most promising individuals for drought and heat tolerance were identified as: 14.KR7.2, 25.X1.1, 35.KR5.5, 36.KR5.6, 41.KR7.5, 46.KR6.4, 47.KR6.5, 291.X1.3, 318.X1.3, and 15.KR7.3. This is the first prospect into the potential of *C. canephora* half-sibs' diversity as an unbound source of genetic variation for abiotic stress tolerance breeding.

**Keywords:** climate change adaptation; *Coffea canephora*; drought tolerance; drought stress recovery; heat stress adaptation; priming by deficit watering

## 1. Introduction

Robusta coffee (*Coffea canephora* Pierre ex A. Froehner) and Arabica coffee are the most globally traded coffee species. *C. canephora* constitutes about 40% of the world's total coffee exports of 117.1 million bags [1]. There are other coffee species gaining attention, such as *C. liberica* var. *excelsa* [2], *C. stenophylla*, and *C. eugnoides*, especially amidst climate change effects [3] and the emergence of specialty markets [4,5]. In the outcrossing diploid ($2n = 2x = 22$) *C. canephora*, the currently underutilized species may vitally be responsible for the crop's diversity and landscape adaptation [6] because of 'unabated' interspecific fertilization among all known coffee species. Of the most commercial species, *C. canephora* is believed to embody high genetic diversity partly attributed to wide geographic adaption [7] and at the plot level, because of the crop's outcrossing behavior [8,9]. Similarly, in some other crops, it is understood that pollination biology has a strong bearing on gene flow and subsequent genetic diversity, e.g., in clonal *Bromelia hieronymi* Mez [10], *Musa acuminata* Colla [11], and coffee-analogous tree species, *Frangula alnus* Mill. [12]. Kiwuka et al. [13] also alluded to similarities in gene pools among *C. canephora* populations in wild, feral,

and cultivated landscapes. In fact, some genotypes of *C. canephora* in Uganda possess morphological architecture (e.g., tree vigor, plant height, leaf color, size, and texture) like that of *C. liberica* as trait similarities enabled by outcrossing (G. Sseremba, pers. comm.). Eight distinct genetic groups of Robusta coffee are known to exist [6,9,10,13–19], including a Ugandan group [13,18].

To further demonstrate the high genetic diversity in *C. canephora*, [13] reported distinct subgroups in Uganda; namely the southcentral (SC) and northwest (NW) clusters. These clusters are quite explained by temperature and drought gradients along the sampled locations [6,13]. In this case, the NW cluster is suggested to have a relatively higher tolerance to drought and high temperature than the SC cluster, whereas geographical location accounts for the wide genetic diversity of *C. canephora* is generally elucidated and planned for utilization [20,21], outcrossing behavior though appreciated [9], is not exploited. Outcrossing or exclusive cross-pollination of the *C. canephora* provides for the random constitution of offspring known as half-sibs of unknown pollen donors but known maternal parentage. In contrast, *C. arabica* withstands inbreeding, thereby supporting pure line selection and reproductively produced commercial seed. There is a need to develop innovative breeding approaches for Robusta coffee with a view to relieving farmers from long reproductive cycles and slow clonal growth progress before access to a commercial variety of guaranteed descriptor composition.

The exploitation of natural half-sibs may be among the most important strategies to shorten a variety's turn-around time in pursuit of meeting market demands arising from climate change effects and rapid changes in customer preferences. Drought and temperature [22,23] stresses among major global coffee productivity and sustainability constraints were selected for piloting the exploration of half-sibs-based genetic diversity in Robusta coffee. According to predictions by [22], an excess in minimum/maximum temperature of 1 °C away from an average of 20.5 °C (16.2–24 °C optimal range) results in at least 14% yield loss of Robusta coffee. Relatedly, drought stress, commonly mimicked under controlled conditions by deficit watering, causes loss of plant turgor, growth retardation, wilting, and ultimate plant death if the stress is unalleviated. Under extreme drought and temperature, yield and bean quality can be impacted to an excess of 80% loss [23].

As demonstrated by [2,13], geographically associated genetic variation exists in relation to the response of *C. canephora* accessions to drought stress. At a variety level, more than one *C. canephora* clone is required to be grown on a farm to maximize clonal diversity, thereby increasing chances of inter-clone pollinations and subsequent fertilization, seed set, and berry development [24]. In addition, studies in other crops have suggested the potential of seed and/or seedling priming in either remodeling the genetic architecture (e.g., through DNA damage repair and new mitochondrial formation) of plants or activating immunity (say, through enzyme activation and protein synthesis) against field constraints so that future stresses would be tolerated or resisted [25–29]. As such, we hypothesized that genetic diversity due to outcrossing behavior might be genotype- and trait-specific depending on the crop seeds' responsiveness to stress adaptation when primed. Plants can respond to drought stress through any or several of the four mechanisms: escape, avoidance, tolerance, and recovery [30–33], but there is no information relating to the impact of deficit watering priming on drought tolerance of Robusta coffee.

This study explores the ability of *C. canephora* to utilize priming effects in half-sib families for protection against future drought and heat stress effects. Through exploration of both drought tolerance and recovery mechanisms under high-temperature environments, we specifically aimed to: (i) ascertain if priming influences tolerance to watering deficit stress; (ii) identify half-sib families that excel under watering deficit stress; and (iii) identify potentially drought tolerant individuals among half-sib families. The intention is to inform on the potential of utilizing genetic diversity created by the outcrossing nature of Robusta coffee in breeding for resilience to drought and high temperature and desired market traits.

## 2. Materials and Methods

### 2.1. Study Site

The study was carried out at the National Coffee Research Institute (NaCORI), Kituza, Mukono, in central Uganda. Kituza is located (latitude −1.406, longitude 34.453) 37 km east of Kampala and 15 km off Kampala-Jinja Road towards the Katosi landing site of Lake Victoria. Mukono district experiences a tropical climate and lies in an agroecological zone of the L. Victoria crescent, characterized by nearly continuous rainfall with two peaks punctuated by moderate dry spells in February and July. Mukono is among low-medium altitude areas at about 1200 m.a.s,l, experiences an ambient temperature range of 16.7–27.8 °C, and is suitable for Robusta coffee production. The overlap in seasonal patterns of the area leads to unreliable flowering patterns, though with peaks (usually in February/March and July/August). Main and fly crop harvest seasons usually occur in November and May, respectively (https://nordicapproach.no/coffee-calendar-2022/ accessed on 14 February 2023). An experiment was set up in controlled conditions of a greenhouse consisting of two contrasting temperature regimes and a rainout shelter. This study was implemented from February 2020 to May 2022. Within Kituza, three temperature-based environments were created and used at test locations, namely general green house (GHG), greenhouse chamber (GHC), and open rainout shelter (ORS) (Scheme 1).

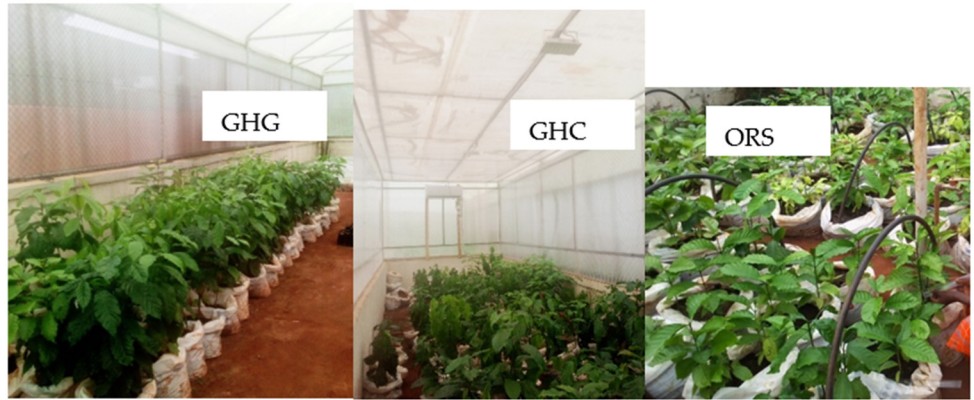

**Scheme 1.** Some of the study coffee half-sib plants under temperature environments of general greenhouse (GHG), greenhouse chamber (GHC), and open rainout shelter (ORS).

The three varying temperature environments included (i) GHG composed of a side metallic chain-link netting and UV-treated carbon polyethene sheet roofing to constitute the mid-day moderate temperature environment of 26–35 °C, (ii) GHC erected inside the GHG to constitute the mid-day high-temperature environment of 32–43 °C, and (iii) ORS which was an open field 'roof-only' rainout shelter without side netting and transparent iron sheets to constitute normal field temperature environment at Kituza for mid-day of 22–27 °C.

### 2.2. Plant Materials

Fully ripe cherry bearing half-sib seed was harvested from three commercial clones of the National Agricultural Research Organisation (NARO) Kituza Robusta (KR) series, namely KR5, KR6, and KR7, and the fourth promising clone coded X1; all of which are known for their resistance to coffee wilt disease, high yield and good physical and cup qualities [34,35]. The cherry seed was then de-pulped to obtain parchment from the coffee cherry. Based on slight modifications of standard protocols by Uganda Coffee Development Authority (UCDA) [36] and World Coffee Research (WCR) [37], freshly de-pulped parchment was sown directly into a substrate consisting of topsoil, sand, and manure in a ratio of 5:2:1 and placed in wooden boxes (Scheme 2). After the deficit watering (priming) treatment at either germination or 4-leaf (seedling) stages, individual plants were transferred to 50-liter high-density (HD) polythene pots for full recovery from priming

stress before re-imposition of deficit watering for experimental pots. Control plants were kept well-watered throughout the experimental period in all the temperature environments.

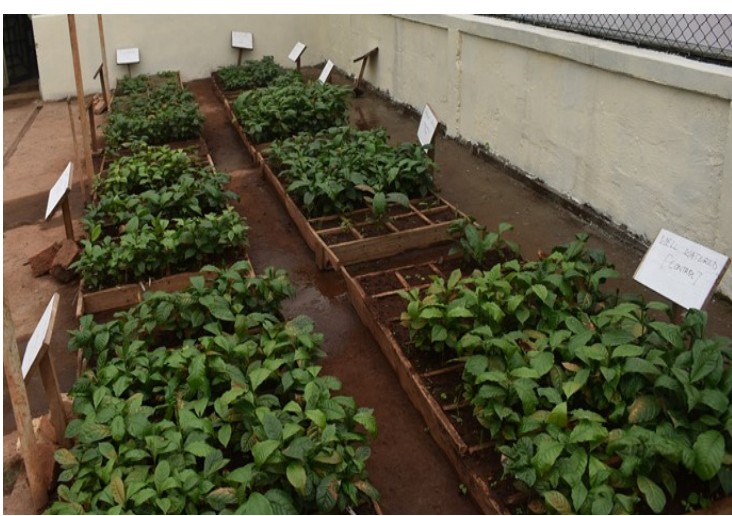

**Scheme 2.** Coffee plants under wooden boxes being recovered through well-watering after half of them had been exposed to deficit watering of 25% field capacity during the germination period.

*2.3. Experimental Set Up*

2.3.1. Design

A nested 'within temperature regime' experimental set up was arranged. Three watering deficit pre-exposure treatments were applied on separate sets (one set for each pre-exposure treatment) of four half-sibs of commercial clones (NARO KR5, NARO KR6, NARO KR7, and a candidate commercial clone X1) of Robusta coffee in each of three temperature environments. Each family was represented by six to 12 plants per irrigation treatment per environment.

2.3.2. Application of Deficit Watering as Priming Method

One of the half-sibs' sets was treated with deficit watering at the germination stage. Similarly, the second set of primed plants was raised under well-watering until the 4-leaf (seedling or early vegetative) stage, at which the deficit watering stress was imposed. The seedlings (1st and 2nd set) under water deficit stress priming (25% field capacity) were later relieved of the stress at nine months of age by well-watering, and this period of well-watering lasted four months, at which age the seedlings had fully recovered. After complete recovery, the primed seedlings were then exposed to watering deficit stress for observation of morphological response on a weekly basis.

A third (control) set of the half-sibs was maintained under well-watering treatment throughout the experimental period. Field capacity (FC) of a growth substrate of 5 topsoil: 2 sand: 1 manure was estimated by volume using a procedure described in [38–40]. At the start of the experiment, twenty-five percent (25%) of FC was used for the priming treatment and 75% for the well-watering treatment, followed by demand-driven adjustments, i.e., based on the age of plants, daily temperature, and relative humidity. During the experiment, we observed that watering at 100% FC was quite excessive, i.e., it brought about flooding conditions in growth pots) and so, 75% was observed to be suited for the controls (well-watered plants).

Adjustments were always based on routine visual observation for wilting symptoms and soil moisture content (SMC) to prevent extreme stress exposure and possible plant death. The SMC was monitored using a digital soil moisture meter (Model MO750 Extech® Instruments Corporation, Waltham, MA, USA) [38]. Other management practices such as weeding, fertilizer application, and pest/disease control were blanketly implemented

throughout the experimental boxes and pots depending on need as per standard recommendations for coffee nurseries and greenhouse practices [36,37].

*2.4. Data Collection*

Data were collected on a fortnightly basis starting from 28 February 2022 to 11 April 2022, every Monday of the week. Ten morphological growth and drought or water deficit response variables were measured following a standard descriptor manual for coffee [41] and applied by [42] with modifications as described in Table 1.

**Table 1.** Description of variables measured on study individuals from different clonal coffee families.

| Variable Abbr. | Variable Name | Unit of Measure | Measurement Procedure |
|---|---|---|---|
| ILP | Internode length on primary branch | centimeters (cm) | Obtained by dividing length of longest/sampled primary by the number of nodes on the primary |
| ILS | Internode length on stem | cm | Obtained by dividing plant height by the number of nodes on stem |
| LBL | Leaf blade length | cm | Leaf blade length was measured from base to apex of a sampled leaf on the first most fully open leaf pair from the primary's growing tip |
| LBW | Leaf blade width | cm | Measured at broadest part of the leaf measured for leaf length |
| LPP | Number of leaves | count | Counted the number of green leaves per plant |
| LLP | Length of primary branch | cm | Measured length of a visually longest primary from its node (point of attachment to the stem) to the farthest lateral growth tip away from the stem. |
| PLH | Plant height | cm | Tape measure was used to record plant height from the collar region to the apical tip of the coffee stem |
| STG | Stem girth | cm | Measured girth of the stem at collar region of the plant using digital vernier calliper at collar region of the potted coffee plant |
| NOP | Number of primaries | count | Counted the number of healthy primary branches on a plant |
| WL | Proportion of wilted leaves | percentage (%) | (No. of wilted leaves/No. of leaves on a plant) ∗ 100 |
| WP | Proportion of primary branches with wilted leaves | % | (No. of primaries with at least one wilted leaf/No. primaries on a plant) ∗ 100 |
| WS | Wilting Score 0+5− | score of 0–5 | wilting score at scale 0–5 (adaptation of Banik et al., 2016: 0 = no leaf is wilted, 1 = 1–25% of leaves are wilted, 2 = 26–50% of leaves are wilted, 3 = 51–75% of leaves are wilted, 4 = 76–100% of leaves are wilted, and 5 = 100% leaf plus stem wilting) |

*2.5. Statistical Analysis*

2.5.1. Priming Effect on Drought Tolerance and Recovery

For drought tolerance study, *F*-test in GenStat 12th edition for a general linear model providing for nesting of replications ($R$) within temperature environments ($E/R$), as well as an interaction of growth stage (weeks, $W$), environment ($E$), priming stage ($P$), and family ($F$), was used as follows:

$$Y_{ijkl} = \mu + E/R_i + W_j + P_k + F_l + EW_{ij} + EP_{ik} + WP_{jk} + EF_{ik} + WF_{jl} + PF_{kl} + EWP_{ijk} + EWF_{ijl} + EPF_{ikl} + WPF_{jkl} + EWPF_{ijkl} + \varepsilon_{ijkl} \tag{1}$$

where $\mu$ is the grand mean, $Y_{ijkl}$ stands for the measured response at the $i^{th}$ replication nested within the environment, $j^{th}$ growth stage, $k^{th}$ priming stage, $l^{th}$ family and all interactions, while $\varepsilon$ is the overall random error term. The decision for discriminability value of a morphological character among families, other factors such as environment and any level of interactions were made at $\alpha = 0.05$.

For analysis of the priming effect on recovery from deficit watering stress, *F*-test at 5% error margin was also used: in this case, considering temperature environment ($E$), replication within the environment ($E/R$), priming ($P$), and family ($F$) treatments. The same levels for each of the study factors as in the drought tolerance analysis were applied for the recovery analysis. In both cases, mean squares for the different sources of variation

and mean values of measured variables per family for control and experimental plants for demonstrating the impact of stress memory are reported.

2.5.2. Identification of Potentially Drought-Tolerant Plants

For both drought tolerance and recovery assessments, the K-means cluster analysis method based on R packages tidyverse for data manipulation [43], cluster for clustering algorithms [44,45], and factoextra for clustering algorithms and visualization (https://cran.r-project.org/web/packages/factoextra/ accessed on 23 December 2022). K-means cluster analysis is appropriate for continuous data [46,47], and it is the case for this study. In general, clustering algorithms are tailored to minimize intra-cluster and maximize inter-cluster variations [45,46]. Of several options for determining the optimum number of clusters based on the location of a bend along a 'total within-cluster sums of squares and the number of clusters (k)' function, the elbow method [45,47] was applied in this study. While analyzing cluster membership, the focus was put on resilience to both water deficit stress (wilting score, WS) and high temperature (growth environment of either GHC or GHG), excluding the ORS at this stage being a relatively low-temperature environment.

## 3. Results

### 3.1. Deficit Watering Effect on Growth and Wilting Response

3.1.1. Family-Level Response

The temperature environment had a significant ($p < 0.05$) effect on all 10 measured variables (Table S1). Plants in open rainout shelter (ORS) had the longest internodes length on primary branches (ILP) at 13.14 cm followed by greenhouse chamber (GHC) at 10.62 cm and general greenhouse (GHG) at 10.22 cm. Priming stage significantly ($p < 0.05$) affected eight of the 10 measured variables, with non-significant ones being the length of internodes on primary branches (ILP) and plant height (PLH). Significant P × F interactions were exhibited for seven of the measured variables except for ILP, the proportion of wilted primaries (WP), and wilting score (WS). The proportion of wilted leaves (WL) for KR5 was lower for priming at germination (14.4%) than at the 4-leaf stage (18.1%) (Table 2). The WL of KR6 and KR7 was also lower for priming at the germination stage than at the 4-leaf stage. It is notable that the KR7 family had the lowest values across priming stages than the rest of the families. These observed trends also hold for WP of KR5 and KR7. However, a reverse trend (compared to that for KR5, KR6, and KR7) for the X1 family was obtained for WL, WP, and WS in that priming at the 4-leaf stage was observed with lower values of wilting than at the germination stage.

**Table 2.** Mean values for growth and wilting traits measured on four different half-sib families of Robusta coffee.

| Variable | Priming Stage | KR5 | KR6 | KR7 | X1 | s.e.d | l.s.d | c.v. |
|---|---|---|---|---|---|---|---|---|
| ILP (cm) | Control | 11.004 | 10.987 | 10.738 | 11.105 | | | |
| | Germination | 11.401 | 10.354 | 11.246 | 11.601 | 0.3952 | 0.7754 | 30.3 |
| | 4-leaf stage | 11.041 | 10.924 | 10.550 | 11.638 | | | |
| ILS (cm) | Control | 5.662 | 5.586 | 5.97 | 5.691 | | | |
| | Germination | 6.399 | 6.093 | 5.976 | 6.099 | 0.1328 | 0.2605 | 19.0 |
| | 4-leaf stage | 5.886 | 5.379 | 6.173 | 6.174 | | | |
| LBL (cm) | Control | 20.994 | 21.735 | 20.74 | 19.807 | | | |
| | Germination | 21.685 | 22.125 | 20.151 | 21.542 | 0.4530 | 0.8887 | 18.1 |
| | 4-leaf stage | 21.809 | 21.038 | 20.848 | 22.664 | | | |
| Leaf blade width (cm) | Control | 9.707 | 10.253 | 9.374 | 8.766 | | | |
| | Germination | 10.07 | 10.307 | 8.927 | 9.817 | 0.2180 | 0.4276 | 19.1 |
| | 4-leaf stage | 9.777 | 9.795 | 9.140 | 10.243 | | | |

**Table 2.** *Cont.*

| Variable | Priming Stage | KR5 | KR6 | KR7 | X1 | s.e.d | l.s.d | c.v. |
|---|---|---|---|---|---|---|---|---|
| LLP (cm) | Control | 23.59 | 20.00 | 25.55 | 24.07 | | | |
| | Germination | 24.82 | 20.30 | 29.81 | 19.17 | 1.330 | 2.609 | 49.2 |
| | 4-leaf stage | 17.58 | 21.14 | 26.64 | 22.30 | | | |
| PLH (cm) | Control | 73.36 | 73.46 | 76.85 | 73.43 | | | |
| | Germination | 74.77 | 73.39 | 73.53 | 70.80 | 1.590 | 3.120 | 18.3 |
| | 4-leaf stage | 74.07 | 71.16 | 73.00 | 76.25 | | | |
| STG (cm) | Control | 1.2028 | 1.134 | 1.2015 | 1.1578 | | | |
| | Germination | 1.1268 | 1.0929 | 1.1839 | 1.0456 | 0.0225 | 0.0441 | 16.9 |
| | 4-leaf stage | 1.0617 | 1.1215 | 1.124 | 1.1059 | | | |
| WL (%) | Control | 0.01 | −0.09 | −0.03 | −0.01 | | | |
| | Germination | 14.35 | 13.48 | 11.30 | 18.99 | 2.127 | 4.173 | - |
| | 4-leaf stage | 18.11 | 18.51 | 14.13 | 13.74 | | | |
| WP (%) | Control | 0.02 | 0.50 | 0.27 | 0.43 | | | |
| | Germination | 32.62 | 33.40 | 31.62 | 35.34 | 2.273 | 4.459 | 84.6 |
| | 4-leaf stage | 35.98 | 32.88 | 36.56 | 34.00 | | | |
| WS (0+5−) | Control | 0.004 | 0.009 | 0.005 | −0.004 | | | |
| | Germination | 1.344 | 1.214 | 1.141 | 1.399 | 0.0979 | 0.1920 | 98.5 |
| | 4-leaf stage | 1.255 | 1.339 | 1.122 | 1.287 | | | |

s.e.d., standard error of differences of means for priming stage by family (P × F); l.s.d., least significant differences of means for P × F at α = 0.05; c.v., percentage coefficient of variation; ILP, internode length on primary; ILS, internode length on stem; LBL, leaf blade length; LBW, leaf blade width; LLP, length of primary branch; PLH, plant height; STG, stem girth; WL, proportion of wilted leaves; proportion of primary branches with wilted leaves; WS, wilting score.

### 3.1.2. Identity of Tolerant Half-Sibs

Either four or eight clusters (Figure 1) were observed to be feasible based on the visible bends in the figure. Subsequent analyses were based on eight clusters (Figure 2). Cluster 7 of size 19 (Table S2) is composed of plants having a relatively low WS of 1.5 and 74% of the individuals located under GHG. The majority (42%) of the cluster 7 members belong to the KR7 family, 26% KR5, while KR6 and XI families are each represented at 16% only. The cluster 7 members include 9.KR6.3, 14.KR7.2, 15.KR7.3, 18.KR7.6, 207.KR5.3, 209.KR5.5, 210.KR5.6, 291.X1.3, 296.KR7.2, 318.X1.3, 320.KR7.2, 25.X1.1, 35.KR5.5, 36.KR5.6, 41.KR7.5, 46.KR6.4, 47.KR6.5, 185.KR7.5, and 243.KR7.3.

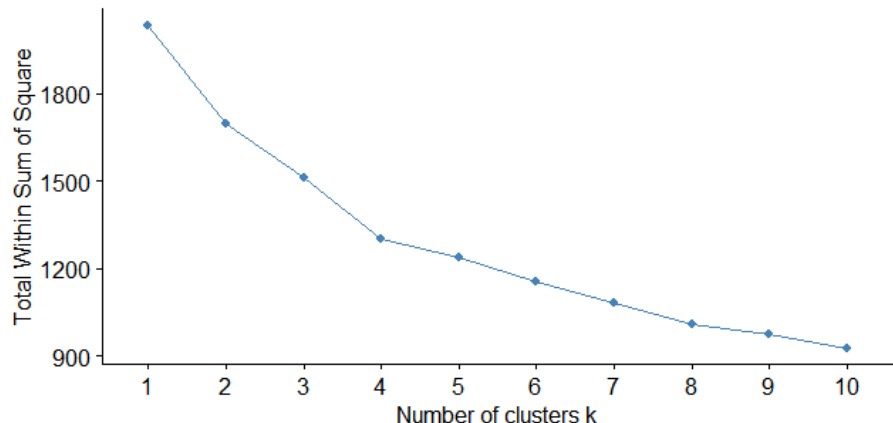

**Figure 1.** Variation of total within-cluster sum of squares with number of clusters (k) indicating optimum k is either 4 or 8 for grouping of Robusta coffee half-sibs grown under contrasting temperature environments and deficit watering.

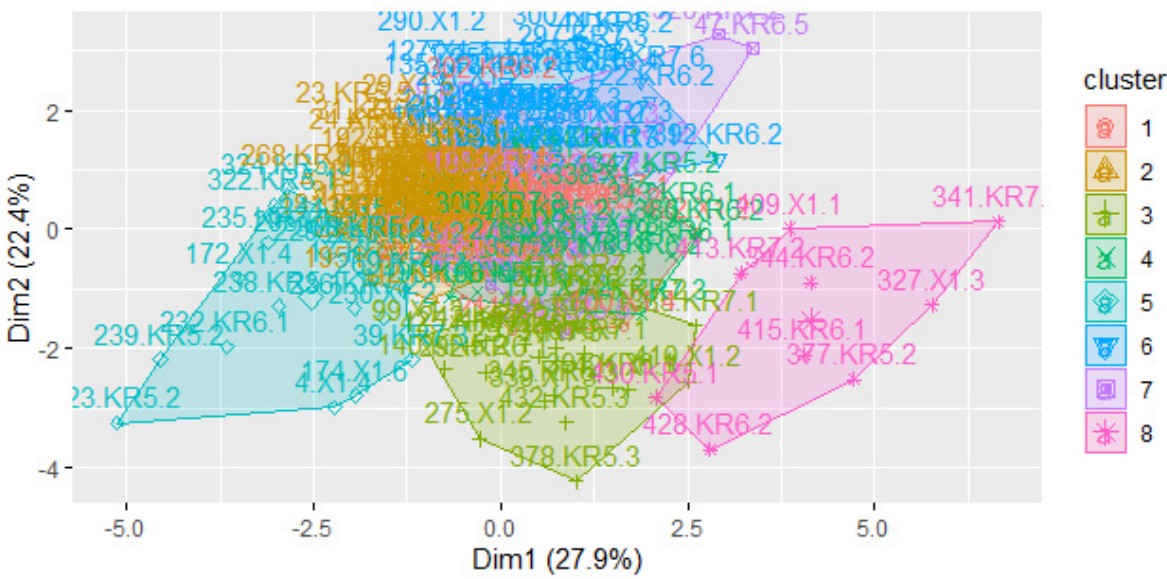

**Figure 2.** Cluster plot for the first two principal components of variation among Robusta coffee half-sibs studied for tolerance to deficit watering.

*3.2. Recovery from Watering Deficit Stress*

3.2.1. Family-Level Response

Temperature environment significantly ($p < 0.001$) affected all 10 measured variables (Table S3), including length of primary (LLP), number of primaries (NOP), and WS. Plants of the longest primary branches were observed under GHC (45.3 cm), followed by GHG (41.2 cm) and ORS (21.3 cm). The highest NOP was recorded under GHC (~7), followed by GHG (~4) and ORS (~3). Aside from the internode length on the stem (ILS), priming significantly affected all measured variables, including LLP, NOP, and WS. There were significant ($p < 0.05$) priming × family (P × F) interactions for internode length on the stem (ILS), leaf blade width (LBW), number of leaves per plant (LPP), PLH, stem girth (STG), and WS. KR6 and KR7 families' values of most of the variables (e.g., LBW, LPP, NOP, PLH, and STG) were higher for plants primed at the germination stage than those at the 4-leaf stage (Table 3). Notably, WS for the K7 family was also lower under priming at the germination stage than at the 4-leaf stage. Overall, however, the lowest WS was obtained with the KR5 family when primed at the 4-leaf stage.

**Table 3.** Mean values for recovery traits measured on four different half-sib families of Robusta coffee.

| Variable | Priming Stage | KR5 | KR6 | KR7 | X1 | s.e.d | l.s.d | c.v. |
|---|---|---|---|---|---|---|---|---|
| ILP (cm) | Control | 9.9 | 10.01 | 9.85 | 9.52 | | | |
| | Seedling | 9.01 | 8.27 | 9.29 | 9.01 | 0.499 | 0.983 | 23.0 |
| | Vegetative | 8.93 | 8.84 | 8.82 | 9.1 | | | |
| ILS (cm) | Control | 6.236 | 6.237 | 6.752 | 6.459 | | | |
| | Seedling | 6.911 | 6.665 | 6.395 | 6.245 | 0.2997 | 0.5901 | 19.9 |
| | Vegetative | 6.25 | 5.767 | 6.28 | 6.563 | | | |
| LBL (cm) | Control | 20.9 | 19.58 | 19.9 | 21.59 | | | |
| | Seedling | 17.75 | 17.63 | 15.44 | 15.81 | 0.956 | 1.882 | 27.7 |
| | Vegetative | 17.81 | 15.56 | 15.33 | 17.43 | | | |
| LBW (cm) | Control | 8.752 | 8.617 | 8.268 | 8.662 | | | |
| | Seedling | 7.443 | 8.012 | 6.488 | 6.492 | 0.4132 | 0.8135 | 23.2 |
| | Vegetative | 7.417 | 6.487 | 6.356 | 7.728 | | | |

**Table 3.** *Cont.*

| Variable | Priming Stage | KR5 | KR6 | KR7 | X1 | s.e.d | l.s.d | c.v. |
|---|---|---|---|---|---|---|---|---|
| LLP (cm) | Control | 27.74 | 23.77 | 28.43 | 26.7 | | | |
| | Seedling | 20.12 | 21.56 | 19 | 17.71 | 3.028 | 5.962 | 59.4 |
| | Vegetative | 19.36 | 16.95 | 19.36 | 18.7 | | | |
| LPP | Control | 53.6 | 48.3 | 53.6 | 54.2 | | | |
| | Seedling | 20.5 | 41.9 | 34.2 | 31.1 | 5.76 | 11.35 | 65.7 |
| | Vegetative | 29.6 | 22.8 | 27.7 | 29 | | | |
| NOP | Control | 7.27 | 5.97 | 7.8 | 6.61 | | | |
| | Seedling | 2.48 | 3.48 | 3.89 | 3.39 | 0.885 | 1.742 | 83.2 |
| | Vegetative | 3.45 | 2.64 | 3.67 | 3.5 | | | |
| PLH (cm) | Control | 84.89 | 85.2 | 88.98 | 86.76 | | | |
| | Seedling | 78.54 | 83.26 | 81.87 | 75.67 | 3.544 | 3.489 | 18.3 |
| | Vegetative | 82.52 | 76.68 | 76.25 | 84.09 | | | |
| STG (cm) | Control | 10.594 | 10.501 | 10.935 | 10.387 | | | |
| | Seedling | 9.973 | 10.222 | 10.644 | 9.844 | 0.411 | 0.810 | 17.0 |
| | Vegetative | 10.064 | 9.691 | 9.339 | 10.734 | | | |
| WS (0+5−) | Control | 0.006 | 0.054 | 0.034 | −0.046 | | | |
| | Seedling | 3.727 | 2.81 | 2.929 | 3.426 | 0.310 | 0.611 | 65.5 |
| | Vegetative | 2.212 | 2.957 | 2.573 | 3.446 | | | |

s.e.d., standard error of differences of means for priming stage by family (P × F); l.s.d., least significant differences of means for P × F at α = 0.05; c.v., percentage coefficient of variation; ILP, internode length on primary; ILS, internode length on stem; LBL, leaf blade length; LBW, leaf blade width; LLP, length of primary branch; LPP, number of leaves per plant; NOP, number of primaries; PLH, plant height; STG, stem girth; WS, wilting score.

### 3.2.2. Identity of Half-Sibs Recovering

Four optimum clusters (Figures 3 and 4) were obtained. Cluster 3, having the lowest WS of 0.9 and size 36 (Table S4), is composed of 72% of its members located under GHG, while the rest (28%) are located under GHC. The majority (33%) of cluster 3 members belong to the KR7 family, followed by 25% X1, 22% KR6, and 19% belong to the KR5 family. The cluster 3 members include 4.X1.4, 8.KR6.2, 12.KR6.6, 13.KR7.1, 14.KR7.2, 15.KR7.3, 16.KR7.4, 18.KR7.6, 25.X1.1, 31.KR5.1, 32.KR5.2, 33.KR5.3, 35.KR5.5, 36.KR5.6, 37.KR7.1, 38.KR7.2, 41.KR7.5, 44.KR6.2, 45.KR6.3, 46.KR6.4, 47.KR6.5, 48.KR6.6, 111.KR7.3, 169.X1.1, 185.KR7.5, 192.KR5.6, 243.KR7.3, 252.X1.3, 276.X1.3, 289.X1.1, 291.X1.3, 292.KR6.1, 297.KR7.3, 309.KR5.3, 312.X1.3, and 318.X1.3.

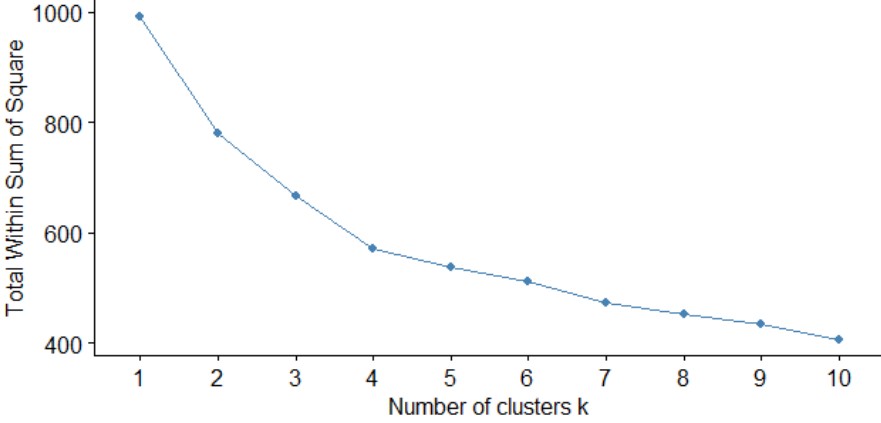

**Figure 3.** Variation in total within-cluster sum of squares with number of clusters (k) indicating that optimum k = 4 for grouping Robusta coffee half-sibs based on drought recovery.

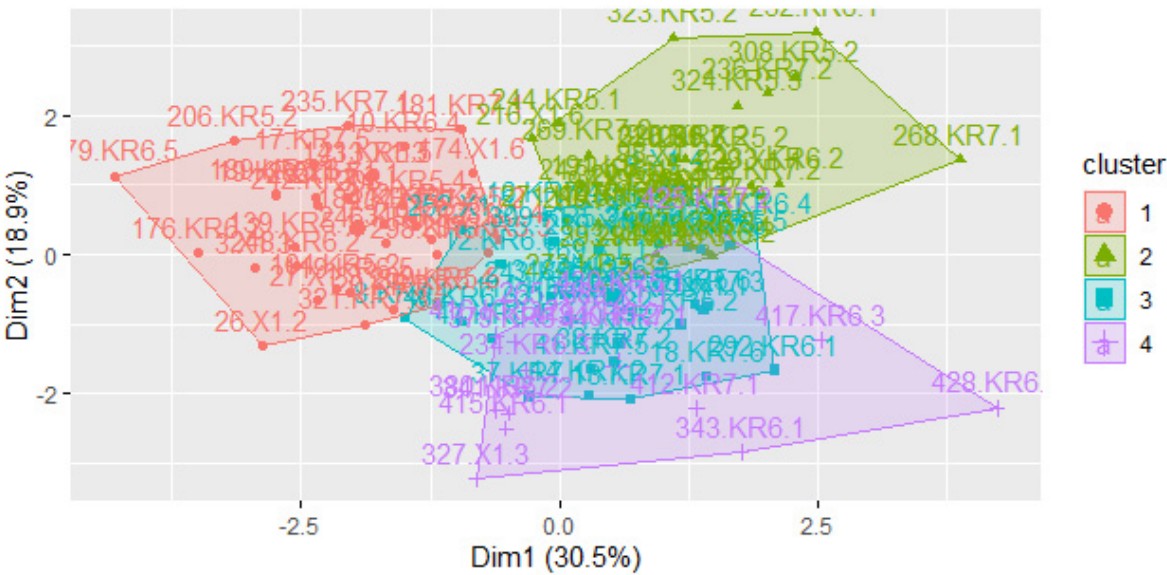

**Figure 4.** Cluster plot for the first two principal components of variation among Robusta coffee half-sibs studied for recovery from watering deficit stress.

Specific examination of data sets on WS for both tolerance and recovery reveals 10 genotypes that appear in both sets and possess between 0 (no leaf is wilted) and 2 (only 26–50% of leaves are wilted), as wilting scores include 14.KR7.2, 25.X1.1, 35.KR5.5, 36.KR5.6, 41.KR7.5, 46.KR6.4, 47.KR6.5, 291.X1.3, 318.X1.3, and 15.KR7.3. The 10 best genotypes were made from high temperature (291.X1.3 and 318.X1.3) and moderately high-temperature environments.

## 4. Discussion

The significance of difference in tolerance to watering deficit stress with stress-primed plants exhibiting better tolerance and recovery than controls suggest a potential of breeding for resilience to drought stress in *C. canephora* using the half-sib selection approach. This study's findings indicate that *C. canephora* populations differ in their ability to acquire tolerance to drought and high temperatures. Individuals, namely 14.KR7.2, 15.KR7.3, 41.KR7.5, and 243.KR7.3, belonging to the KR7 family, posed a relatively greater potential in tolerating simulated drought/water deficit and high temperature than the other families. Similarly, the greatest potential for recovery from drought stress was exhibited by individual plants of the KR7 family, i.e., 14.KR7.2, 15.KR7.3, 41.KR7.5, 185.KR7.5, and 243.KR7.3. For both tolerance and recovery, potential genotypes for drought tolerance are indicated as 14.KR7.2, 25.X1.1, 35.KR5.5, 36.KR5.6, 41.KR7.5, 46.KR6.4, 47.KR6.5, 291.X1.3, 318.X1.3, and 15.KR7.3. Although at the family average level, KR7 responded better to drought tolerance and recovery, indicating a possible role of maternal genetic factors [33,48]; the 10 best individual plants (genotypes) are a representation of each family studied. Of the best 10, three come from KR7 (14.KR7.2, 41.KR7.5 and 15.KR7.3), three from X1 (25.X1.1, 291.X1.3 and 318.X1.3), two from KR5 (35.KR5.5 and 36.KR5.6), and two from KR6 (46.KR6.4 and 47.KR6.5). Our view is that the observed distribution of best genotypes is a result of (i) the genetic background of mother clone KR7 being more drought tolerant than that of the rest of the studied materials, but (ii) there is also a random pollen grain movement due to an exclusive cross pollination tendency in Robusta coffee thereby increasing geneflow among the study half-sib offspring.

The random mating facilitated by self-incompatibility of *C. canephora* [24,49] and pollination agents [24,50] creates unlimited genetic diversity that suites the crop to recurring climate change effects (biotic and abiotic) [2,22,23], including the studied watering deficit tolerance and recovery from the stress. Thus, some of the recombinant genetic compositions from outcrossing tendency could have occurred and conditioned differential

stress memory abilities, especially when exposure occurred at the germination stage. This resilience development technique is popularly known as priming, and positive results have been realized in other crops [25,28,29]. The ability of crop plants to adapt to abiotic stresses is often influenced by a combination of genetic and environmental factors. The significance of any of these factors, in turn, determines the heritability and durability of desired attributes in a genotype. Genetically, a plant may possess active tolerance factors outright; otherwise, activity has to be initiated early in the development stage by way of the plant's innate immunity activation or enforcement of adaptation through artificial means such as crossbreeding and backcrossing, mutation breeding, and genetic engineering. The coffee sector is not yet ready for genetic engineering. Thus, an option of stimulation of remodeling in genetic architecture by stress pre-exposure is imminent, especially in the case of Robusta coffee breeding that is constrained by unpredictable flowering time, making synchronization for parents difficult [24,50], long reproductive cycles, and slow clonal propagation [51,52]. During artificial immunity activation or induction, a plant's response may manifest and qualify as either being an escapee, avoidant, tolerant, or recoveree. The ability and level of response, as well as the acquired mechanism of drought tolerance, is understood to be both genotype and environment dependent [7,14,23,31].

Our findings generally indicate better watering deficit stress tolerance and recovery for plants primed at germination than at the 4-leaf stage. The views held based on the findings are subject to a follow-up validation study. The validation involving both mother clones and the promising half-sibs needs to be undertaken beyond the controlled conditions of the greenhouse. This can demonstrate the use of half-sibs' unlimited genetic constitution and diversity for selecting varieties amenable to the increasing effects of climate change that risk on-farm sustainability.

## 5. Conclusions

Watering deficit pre-exposure improves subsequent tolerance to the stress in Robusta coffee half-sibs; maternal origin or family of the half-sib influenced the response to priming stress. KR7 family was better than others in tolerance to deficit watering stress. Selections recommended for onward use in breeding for drought and high-temperature tolerance include 14.KR7.2, 25.X1.1, 35.KR5.5, 36.KR5.6, 41.KR7.5, 46.KR6.4, 47.KR6.5, 291.X1.3, 318.X1.3, and 15.KR7.3. Four of the selections (14.KR7.2, 291.X1.3, 318.X1.3, and 15.KR7.3) are products of stress pre-exposure at the germination stage. We observed that genetic diversity due to outcrossing behavior is genotype- and trait-specific depending on the crop seeds' ability to respond to priming at early development stages.

**Supplementary Materials:** The following supporting information can be downloaded at: https://www.mdpi.com/article/10.3390/agronomy13030674/s1, Table S1: Mean squares (MS) and significance levels (p) for rejecting a hypothesis of no difference in morphological response among priming stages and Robusta coffee families; Table S2: Cluster vector and attributes of Robusta coffee half-sibs under deficit watering conditions; Table S3: Mean squares (MS) and significance levels (p) for rejecting a hypothesis of no difference in recovery response among priming stages and Robusta coffee families; Table S4: Cluster vector and attributes of Robusta coffee half-sibs based on recovery from deficit watering stress.

**Author Contributions:** Conceptualization, G.S., P.B.T., P.M., J.S.Y.E., G.H.K., E.B.K. and G.A.; Methodology, G.S., P.M. and J.M.; Validation, L.D.M., D.N.B., G.H.K., E.Y.D. and G.K.; Formal analysis, G.S., L.D.M. and N.A.; Investigation, G.S., E.A., J.M., N.A., E.M., J.K., B.M. and G.H.K.; Resources, P.B.T., E.Y.D., E.B.K., G.K. and E.I; Data curation, D.N.B., E.A., N.A. and E.M.; Writing—original draft, G.S.; Writing—review and editing, P.B.T., P.M., J.S.Y.E., L.D.M., D.N.B., E.A., J.M., J.K., B.M., E.B.K., G.K. and E.I.; Supervision, P.B.T., P.M., J.S.Y.E., E.Y.D., E.B.K. and G.A.; Funding acquisition, G.S., E.Y.D., G.K. and G.A. All authors have read and agreed to the published version of the manuscript.

**Funding:** This research was funded by TWAS/UNESCO and Sida grant number 20-354 RG/BIO/AF/AC_G—FR3240314179 for research equipment, DAAD under the Postdoctoral Fellowships in Sub-Saharan Africa at DAAD supported Centres grant number 91817875 and Uganda Coffee Development Authority (UCDA). The APC was funded by UCDA.

**Data Availability Statement:** The data used in article is available on request and we can upload it online for access as per data policies.

**Conflicts of Interest:** The authors have no conflict of interest to declare.

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
