# Peer review of "Viability of Deficit Irrigation Pre-Exposure in Adapting Robusta Coffee to Drought Stress"

_agronomy, doi:10.3390/agronomy13030674_

Round 1
Reviewer 1 Report
Drought is one of the major global threats to coffee production. In this manuscript, the authors explored the ability of C. canephora to utilize immunity inducement in half-sib families for protection against future drought stress effect. In my opinion, this manuscript can be accepted after revision.
1. Line 162-163, for‘5topsoil:2sand:1manure’, some blanks should be inserted.
2. Table 2 and 3 should be combined and move to supplementary table.
3. The quality of all figures should be improved, especially for resolution.
4. For Table 5 and Table 9, the detail member information for each cluster can be move to supplementary table.
5. How did the authors obtain the values of variables for table 5 and 9, the average ? If the average values were used, the standard error should be added.
6. Table 6 and 7 should be combined and move to supplementary table too.
7. For Table 4 and 8, the column of variable can be replaced by the variable abbreviation that defined in Table 1. And the mean value should be expressed as mean value ± standard error.
Reviewer 2 Report
Major concerns
1. The authors should provide some pictures of the environments, variables and morphological performance under treatments in methods, which will lead a better understanding for readers.
2. The collected data should be listed in supplementary materials.
3. Has the authors examined the temperature change in a whole day under the three environments? Which is necessary to known how much time of the high temperature was lasted under 35°C and 43°C.
4. What are the meanings of the number before and after the name of varieties? Which should be explained.
5. The discussion should be rephrased. Plant’s innate immunity is commonly related with biotic stress. Has its relation with abiotic stress been clearly revealed? The sentences with genetic factors should be more cautious since there is no genetic experiment conducted in the manuscript.
6. How was the ‘stress memory’ concluded in the title and conclusion? Which should be obviously pointed in result and discussion.
Reviewer 3 Report
Eventually, the work can be important, but clear exposition, with fluing ideas, together with clear English, are missing on many parts of the manuscript
Title:
- Must be improved, giving the clear idea about the experiment results. Memory is not the real focus of your work.
Abstract:
- We are not interested in statistical significance, but in execution, results, and recommendations.
- Please, read details in pdf.
Introduction:
- Thoughts and ideas are not clear and focused.
- The story about coffee species must be concentrated and well exposed.
- Did you work with seeds because your hypothesis seems that you did?
- The immunity had not been introduced and was the focus of your work. Necessary to improve!
- There are papers about memory of stresses in coffee: Menzes-Silva et al., 2017; Rakocevic and Batista, 2020; DaMatta et al., 2021. Please, introduce the concept of memory to stresses.
- Clean the introduction about histories without links to your experiment.
- The references are written wrongly, please follow the template.
- English is somehow very difficult to understand, verbs are missing, or unusual words are used.
- Please, read details in pdf.
M&M:
- The climate is tropical, the classification, please? What is maximal/minimal temp?
- Give us the coordinates.
- Harvest of what crop?
- Berries were pulped by wet processing, or de-pulped? Badly written.
- Did you sow parchment or seeds?
- You are talking about priming and recovery priming (Lines 138-144), but it is very confusing. Please, explain better.
- The drought stress is not clear at all, watering no more. You must be explicit: At day… such water quantity, such frequency… How was the water applied? Manually?
- How did you measure the soil water potential?
- How the experiment of three-four months and seedlings were watered well after nine months? Please, be careful and coherent.
- What does it mean visual observation for adjustments (line 168)?
- - What is primary, what is stem? Please, read about the C. canephora architecture (Rakocevic et al., 2022, Tree Physiology, DOI: 10.1093/treephys/tpac138) and use the correct, scientific terms, not technical field jargon.
- The table 1 can be explained as the simple text. Nothing new, or no data was exposed.
- Any program is providing a general linear model. What program, or software, did you use? What package?
- Put ‘tidyverse’… for package…
- Too much and too nothing about statistics. If you are developing one specific statistical methos, please, denote this in Introduction. If not, you do not need so many explanations in M&M. Be direct, explain well, because there are too much abbreviations, but essential had not been explain, the protocol of soil drying, soil water potential and plant responses.
Results:
- Very complex model, with four factors. It is not at all easy to follow. What is m.s, F. pf? Please, write the title of Table explicitly.
- Means square is not a data useful for biology, or agronomy. Please, put mean values.
- I prefer that you improve your model to 2-factorial, or max, 3-factorial, while time can be observed as one independent factor.
- In table, all abbreviations about factors, must be explained. No reader has the elephant memory.
- Three immense tables and no results from them. You must repeat your stats and give some results. Means, what did change. Your results in three tables, in a present form, are useless.
- Figures 1 and 2 are useless, too. They must give some information, including variation and errors. There is no result, only charts. What is the meaning of those colored clusters. No legend is done.
- How did you come to results of incredibly long tables 5 and 9?
- One useful recommendation: Pool statistics and means, make useful tables and charts. The extensive tables put into supplementary material. Be direct, write about your results, because you understand what you did, and you are transmitting to others.
Discussion:
- Please, you must grow for discussion, for all interactions and comparisons. Please, read results of other scientists.
Conclusions:
- You are talking about repetitive drought expositions, but your experiment was not explained and give the impression that the drought was constant. Please, return, explain everything.
- If your M&M and Results were OK, and they are not, the best part of your manuscript would be 'Conclusions'.

Reviewer 4 Report
Dear Authors!
The article is interesting. You have done a hard job.
My suggestions:
1) In Abstract you should add methods of stress tolerance evaluation. For example, ''... later evaluated for tolerance to water deficit in three different temperature environments by estimation of plant growth and wilt parameters.'' Add the order in which resistant clonal families are listed (decreasing stress-tolerance?).
2) Decipher for the first time: UCDA, WCR (page 3)
3) I didn`t understand words primary and tidyversa (page 5). Is primary a common term? Is it seedling? I saw this term in the context of primary cell wall and primary metabolism of plants.
4) Check the text again. For instance "...six to 12 plants..." (page 3), "... eight of 10 measured..." (page 5), "... reveals10 genotypes..." (page 13).
5) Reference should be described according instructions for authors.
Reviewer 5 Report
Overall this manuscript is addressing a relevant aspects related to Coffea canephora and its potential for tolerance of watering deficit caused dry periods. The manuscript follows all elements for a scientific article. However, some changes need to be done for the improvement of its quality.
I believe some aspects in results and discussion need to be re-write for make it easer for readers to understand. The topic is complex and authors are making it even more complex in the way results and discussion are presented. See attached PDF file.

Round 2
Reviewer 1 Report
Now this manuscript is much more readability. In my opinion, this manuscript can be accepted with some small revisions.
1. Line 212, 215, and other places: ‘significantly (p<0.05)’ should be ‘significantly (p<0.05)’. The letter ‘p’ should be italic.
2. For the "Conclusion section", please write only the concluding remarks. Already, much is described in the discussion section.
Reviewer 2 Report
I still insist that the authors should provide some pictures of the environments, variables and morphological performance under treatments in methods, which will lead a better understanding for readers.
Reviewer 3 Report
- - I was extremely disappointed with the reaction of researchers. They made some improvements, but one part of suggestions they did not respect. I did not receive any feedback, nothing that prove any of mine suggestions and generally, they had not been respected.
- Details can be follow in 'pdf' of the manuscript.
Title:
- Was improved
Abstract:
- Was improved
Introduction:
- Was improved, but many references are still written by names. And text is not sufficiently sophisticated.
M&M:
- The climate is tropical, the classification, please? What is maximal/minimal temp? How much rains and their distribution? How can you talk about elevated temperatures and they are not explicit?
- The coordinates are done in unusual way, please, make them clear, without necessities of million digitals.
- No explanation was improved.
- How was the repetitive drought exposure despaired???????
- No data about drought in water potential is done.
- No data about well-watering is done. What does it mean????
- The drought stress is not clear at all, watering no more. You must be explicit: At day… such water quantity, such frequency… How was the water applied? Manually?
- How did you measure the soil water potential?
- How the experiment of three-four months and seedlings were watered well after nine months? Please, be careful and coherent.
- The table 1 can be explained as the simple text. Nothing new, or no data was exposed.
- You are researchers. Please, use botanical definitions of Hallé et al., 1978, as done in C. canephora architecture (Rakocevic et al., 2022, Tree Physiology, DOI: 10.1093/treephys/tpac138) and use the correct, scientific terms, not technical field jargon. Stem = orthotropic, Primary = plagiotropic branches of 2nd order.
Results:
- Tables are the simple list of means! No comparison, no p-value, no statistical error, no ranking, no grouping.
- Figures 1 and 3 are useless in the manuscript. Or 2 and 4. Never mentioned!!!!!!!! How did you make clusters, when, what is the meaning????
- This is definitively not one mature work. Please, work on presentation. Make clear to yourself your work, and after that to us.
Discussion:
- Please, you must grow for discussion, for all interactions and comparisons. Please, read results of other scientists.
Conclusions:
- How was the repetitive drought exposure despaired???????
- as in the previous version - if your M&M and Results were OK, and they are not, the best part of your manuscript would be ‘Conclusions’.
References:
- Not in according to MDPI rules.

Reviewer 4 Report
Dear Authors!
I have only one small recommendation: put the numbers of references on page 2 in square brackets.
Reviewer 5 Report
This manuscript was improved.
Author Response
Thank you very much for the quick and supportive review!